# MEMS-Based Reflective Intensity-Modulated Fiber-Optic Sensor for Pressure Measurements

**DOI:** 10.3390/s20082233

**Published:** 2020-04-15

**Authors:** Ning Zhou, Pinggang Jia, Jia Liu, Qianyu Ren, Guowen An, Ting Liang, Jijun Xiong

**Affiliations:** Science and Technology on Electronic Test and Measurement Laboratory, North University of China, Taiyuan 030051, China; zhouningnuc@163.com (N.Z.); 18734920710@163.com (J.L.); renqianyu1315@126.com (Q.R.); anguowen@nuc.edu.cn (G.A.); liangtingnuc@nuc.edu.cn (T.L.); xiongjijun@nuc.edu.cn (J.X.)

**Keywords:** fiber-optic, reflective intensity modulated, MEMS, gas pressure sensor

## Abstract

A reflective intensity-modulated fiber-optic sensor based on microelectromechanical systems (MEMS) for pressure measurements is proposed and experimentally demonstrated. The sensor consists of two multimode optical fibers with a spherical end, a quartz tube with dual holes, a silicon sensitive diaphragm, and a high borosilicate glass substrate (HBGS). The integrated sensor has a high sensitivity due to the MEMS technique and the spherical end of the fiber. The results show that the sensor achieves a pressure sensitivity of approximately 0.139 mV/kPa. The temperature coefficient of the proposed sensor is about 0.87 mV/°C over the range of 20 °C to 150 °C. Furthermore, due to the intensity mechanism, the sensor has a relatively simple demodulation system and can respond to high-frequency pressure in real time. The dynamic response of the sensor was verified in a 1 kHz sinusoidal pressure environment at room temperature.

## 1. Introduction

Pressure measurement in harsh environments is of great value in various fields, such as oil logging, aerospace vehicle engine testing, and in the pharmaceutical industry [1,2,3]. At present, pressure sensors used in engineering mainly include piezoresistive pressure sensors, piezoelectric pressure sensors, capacitive pressure sensors, and fiber optic pressure sensors [4,5,6,7]. Compared with the electrical sensor, the fiber optic sensor has unique advantages. Fiber optic sensors are more adaptable to harsh environments, more resistant to electromagnetic interference, and have compact size and simple structure [8]. Until now, fiber-optic pressure sensors have been developed into many subtypes according to the working mechanism, including modulating intensity [9,10], frequency [11], phase [12], wavelength [13], and polarization [14].

In recent years, intensity-modulated fiber-optic sensors have attracted significant interest due to their simple system, low cost, and high dynamic response. Shen et al. proposed a fiber-optic displacement sensor, which is based on reflective intensity modulation using a fiber-optic collimator [15]. Vallan et al. utilized a low-cost plastic fiber-optic sensor for displacement and acceleration sensing, and verified it under the actual conditions of a sinusoidal vibration [16]. Perrone et al. reported a novel non-contact method to measure vibrations, which enabled a submicron resolution [17]. Cui et al. designed a new dual fiber structure sensor based on a fiber Bragg grating probe that can measure the axial and radial contact displacement simultaneously and can be used to measure the shape change of the fuel nozzle in engines [18]. Vallan et al. proposed and analyzed an intensity-modulated sensor for 2D crack monitoring [19]. Guermat et al. presented a fiber-optic sensor that could monitor the temperature and pressure. The measurements were taken by a reflectometer [20]. Due to the limitations of temperature-resistant materials and the current processing technologies, the above sensors are not suitable for a harsh temperature environment or batch-production. 

With the development of technology, a considerable number of methods have been applied to fabricate the fiber-optic sensors [21,22,23], for example, MEMS, chemical etching, lasing, applying special optical fibers, and so on [24,25,26,27,28]. Ge et al. proposed an optical MEMS pressure sensor based on a mesa-diaphragm structure [29]. Poeggel et al. reported a femtosecond-laser-based inscription technique for post-fiber-Bragg grating inscription in an extrinsic Fabry-Perot interferometer pressure sensor; the sensor has a small size and high stability [30]. Hirsch et al. reported the fiber-optic microsphere sensors, which realized high-sensitivity refractive index detection [31]. The MEMS-based optical fiber pressure sensors have attracted significant interest, as they are quite small and ideal for applications where restricted space or minimal measurement interference is a consideration. 

In this study, we proposed a MEMS-based reflective intensity-modulated fiber-optic sensor for pressure measurement. The sensor consists of two multimode optical fibers with a spherical end, a quartz tube with dual holes, a silicon sensitive diaphragm, and a high borosilicate glass substrate (HBGS) integrated by MEMS technique. The sensor was assembled and sealed using a CO2 laser, which is beneficial for improving the sensor performance and avoiding thermal mismatch between the adhesive and the fiber-optic. The temperature features were characterized over the temperature range of 20-150 °C, and the dynamic response was verified at room temperature. The sensitivity of the sensor was significantly improved by fabricating a micro-sphere at the end of the multimode fiber. Due to the intensity mechanism, the sensor has a relatively simple demodulation system and can respond to high frequency pressure. Besides, the pressure sensor proposed in this paper has the potential to be mass-produced, which can reduce manufacturing costs.

## 2. Configuration and Operating Principle

The configuration of the proposed sensor is shown in Figure 1a. The sensor consists of two multimode fibers with a spherical end, a quartz tube with dual holes, and a sensor head. The sensor head structure consisted of an HBGS covered with a silicon sensitive diaphragm. Two multimode optical fibers were inserted inside the quartz tube and further vertically into the sensor head. A pressure cavity was formed between the ending of the optical fiber and the inner surface of the silicon sensitive diaphragm. The sensing principle of pressure is shown in Figure 1b. The light was transmitted in the transmitting fiber and incident on the sensitive diaphragm, then reflected by the surface of the sensitive diaphragm, and received by the receiving fiber. The final reflection spectrum was analyzed by a photodetector. When an external pressure was applied to the sensitive diaphragm, the diaphragm deformed, which induced the optical path change. The output voltage of the photodetector changes linearly with the pressure. By processing the voltage signal by the photodetector, the change in the external pressure can be monitored in real time. 

When the light propagates in the transmitting fiber, the radial distribution along the fiber axis can achieve an approximately Gaussian distribution. The irradiance of emitted light from the transmitting fiber obeys an exponential law according to [32,33,34]:
(1)$$I(r,z)=\frac{2P_{0}}{\pi w^{2}(z)}e^{\frac{-2r^{2}}{w^{2}(z)}},$$
where $P_{0}$ is the optical power sent by the transmitting fiber; r and z are the radial coordinates and pressure cavity length, respectively; and w(z) is waist radius of the reflected light at the sensitive diaphragm and can be defined as:(2)$$w(z)=w_{0}+2ztan(acrsin(NA)),$$
where $w_{0}$ is the radius of the transmitting fiber, and NA is numerical aperture of the transmitting fiber.

The receiving fiber collects the light reflected from the sensitive diaphragm, and the total received optical power $\left(P_{z}\right)$ can be evaluated by integrating the irradiance I(r, z) within the core region ($S_{r})$ of the receiving fiber end:(3)$$P_{z}=\int_{S_{r}}I(r,z)dS_{r}.$$

To better describe the light intensity modulation characteristics of the sensor, the light intensity modulation function (M) is defined as follows:(4)$$M=\frac{P_{z}}{P_{0}}=F(w_{0},w_{r},s,NA,z),$$
where s is center-to-center distance between two fiber cores, and $w_{r}$ is the core radius of the receiving fiber. The designed parameters of the proposed fiber-optic pressure sensor obtained by Equation (4) are listed in Table 1. 

Compared with the standard fiber, the fiber with microsphere structure has a higher numerical aperture (NA) [35]. The relationship between the NA of the optical fiber and the light intensity is simulated using the MATLAB software, as shown in Figure 2. The initial pressure cavity length of the sensor is $z_{0\;}\;(z_{0}>s+w_{r})$. In the effective working range [36], the sensor with higher numerical aperture fibers has a larger slope (k) in the intensity modulation function $\left(k_{n_{3}}>k_{n_{2}}>k_{n_{1}}>k_{n_{0}}\right)$. Thus, by using the fiber with a microspherical end, the pressure sensitivity of the sensor could be effectively improved. 

## 3. Fabrication of the Sensor

A 4-inch double-side polished silicon wafer was used to manufacture the sensitive diaphragm. The HBGS was made by a 4-inch high borosilicate glass. The thicknesses of the silicon and high borosilicate glass were 300 μm and 2 mm, respectively. The whole manufacturing process mainly included silicon sensitive diaphragm processing, HBGS processing, sensor head assembly, and fiber integration. The fabrication process is as follows: Firstly, a mask layer of the photoresist was coated on one side of each silicon wafer. Secondly, the silicon wafer was subjected to standard lithographic processes using a pattern with a diameter of 3 mm. After this, dry etching was applied to form a deep cylindrical cavity with a diameter of 1.5 mm and a depth of 210 μm, as shown in Figure 3a–c. The diaphragm, with a final thickness of approximately 90 μm, can enhance the pressure sensitivity. Subsequently, the photoresist was repeatedly applied on the back surface, and a second photolithography process was operated to form a back pattern with a diameter of 3 mm. Then, a gold film was formed on the silicon wafer using the magnetron sputtering technique, and the rest photoresist was removed. The silicon sensitive diaphragm was made by using the above steps, as shown in Figure 3d–g. Due to the high reflectivity of Au, the optical coupling efficiency of the sensor can be improved.

After this, a micro-machining method was used to fabricate a cylindrical deep cavity with a diameter of 3 mm and a depth of 0.3 mm on one side of a high borosilicate glass. On the other side of the high borosilicate glass, a convex platform with an outer diameter of 2 mm and a height of 1 mm was processed. A 1mm-diameter hole was machined at the center of the entire structure. The HBGS could be successfully processed, as shown in Figure 3h. The HBGS was then anodically bonded with the silicon sensitive diaphragm. Then, the entire sensor head was assembled by using the above steps, as shown Figure 3i. For the fiber integration, we inserted two multimode fibers (MMF125/62.5, YOFC, Wuhan, China) with the microsphere end into a quartz tube. We then inserted the entire tube into the sensor head vertically. The microsphere structure was made by laser heating at the end of the fiber. The carbon dioxide laser fusion splicer (LZM-110, Fujikura, Ltd., Tokyo, Japan) was used in the production process. The laser power was 175 bits (about 5.2 W), the laser heating time was 2 s, and the heating was performed twice [37]. The sensor was assembled and sealed by using CO2 laser, which was beneficial for improving the sensor performance and avoiding thermal mismatch between the adhesive and the fiber-optic [38], as shown Figure 3j. The length of the initial pressure cavity is 190 μm.

The entire pressure sensor was 5.0 mm in length, 5.0 mm in width, and 1.3 mm in height, as shown in Figure 4. Figure 4a shows the real image of the sensor. The microsphere end of the fiber was observed under a microscope, as shown in Figure 4b,c. Figure 4d provides a sectional view of the double-hole quartz casing with an outer diameter of approximately 1 mm and an aperture of approximately 126 μm.

## 4. Experimental Results

The experimental setup of the sensor for pressure testing under the dynamic temperature is shown in Figure 5, which includes the fiber sensing system and the temperature and pressure control system. The fiber sensing system consisted of a LED light source (850 T, Wyoptics, Shanghai, China), a fiber optics coupler (WLF 1 × 2 MM ratio 50:50, 850nm, HJGTEK, Shenzhen, China), a photodetector (New Focus Model 2001, New Port, California, America), and an oscilloscope (TBS 1102, Tektronix, America). The photodetector was set as Gain knob:$\;10^{4}$, Gain multiplier: 3, and Response factor: 0.36. The temperature and pressure control system includes a thermal chamber, a pressure vessel, a gas cylinder, a temperature, and pressure controller. 

During the experiment, the sensor was placed in the thermal chamber. A thermocouple was placed near the sensor, the value of which was displayed on the temperature controller. The pressure controller can ensure a uniform distribution of pressure in the pressure vessel. First, at room temperature, we increased the pressure from approximately 0 to 1 MPa at 0.1 MPa steps to verify the voltage response. The pressure was kept for 5 min at each step, and the corresponding output voltage of the sensor was recorded. Figure 6 shows the fitting curves of the voltage versus pressure during the three pressurization experiments. A linear relationship between the voltage and pressure was observed. The pressure sensitivities of the three tests were 0.1391, 0.1390, and 0.1390 mV/kPa, respectively. For these three repeatable experiments, the repeatability and nonlinear errors are approximately 2.15 % and less than 2.51 %, respectively.

To test the temperature performance of the sensor, we monitored the pressure response of the sensor under 20 °C, 50 °C, 75 °C, 100 °C, 125 °C, and 150 °C, respectively. Figure 7 shows the response voltage under the different temperatures, and reveals that the response to the pressure at different temperatures is linear. The pressure sensitivities at temperatures of 20 °C, 50 °C, 75 °C, 100 °C, 125 °C, and 150 °C, were 0.1387, 0.1409, 0.1431, 0.1452, 0.1460, and 0.1473 mV/kPa, respectively. It was seen that the temperature had a small effect on the sensitivity of the sensor, which may be caused by complex physical mechanisms such as the Young’s modulus of the sensitive diaphragm materials changed with the temperature. Although the temperature has a certain influence on the sensor, it still showed a good response to the pressure in a temperature environment and achieves good linearity. Figure 8 shows the relationship between the initial sensor voltage with temperature. The experiment results show that the temperature coefficient of the proposed sensor was 0.87 mV/°C. When the temperature exceeded 150 °C at a pressure of 1 MPa, the welding part of the sensor was damaged, resulting in the pressure cavity of the sensor not being sealed. Therefore, the sensor can operate at a higher temperature than 150 °C, when the welding process is optimized.

To verify the dynamic response of the sensor, a dynamic pressure test system was established, as shown in Figure 9. It consisted of a pressure control system, a sinusoidal pressure generator, a standard piezoelectric sensor, a gas cylinder, a pressure vessel, and a fiber sensing system. Standard piezoelectric sensors were used to monitor pressure changes and feed them back to the pressure control system. During the dynamic experiment, the sensor head was placed in the pressure vessel. The pressure control system controlled the magnitude and frequency of the pressure applied to the sensor. To verify the pressure response of the sensor under dynamic conditions, experiments at 400 Hz, 780 Hz, and 1 kHz were performed. Figure 10 shows the output of the sensor under sinusoidal pressure environments at room temperature. According to the experiment results, the sensor had a good dynamic response under 400 Hz, 780 Hz, and 1 kHz sinusoidal pressure environments. 

## 5. Conclusions

We proposed and demonstrated a reflective intensity-modulated fiber-optic pressure sensor that can be applied under various pressures from 0 to 1 MPa, at room temperatures up to 150 °C. The sensor can also be applied in a dynamic pressure environment and monitor a high-frequency dynamic pressure signal in real time. We used an optical fiber with a microspherical end, which effectively improved the pressure sensitivity of the sensor. The whole sensor was integrated and sealed with a CO_2_ laser, which is beneficial for improving the sensor performance and avoiding thermal mismatch between the adhesive and the fiber-optic. Moreover, the sensor is fabricated by MEMS techniques, providing the possibility of the batch production. 

## Figures and Tables

**Figure 1 sensors-20-02233-f001:**
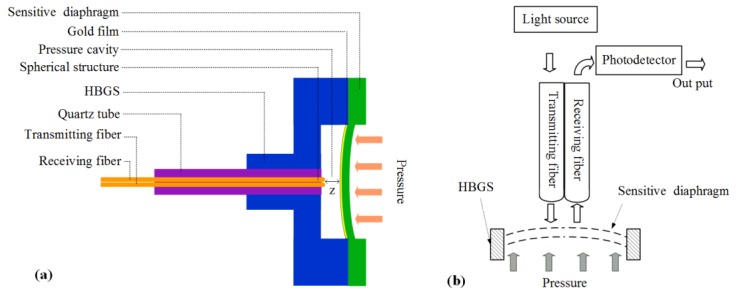
Reflective intensity-modulated fiber-optic pressure sensor: (**a**) structural configuration; and (**b**) principle of pressure sensing.

**Figure 2 sensors-20-02233-f002:**
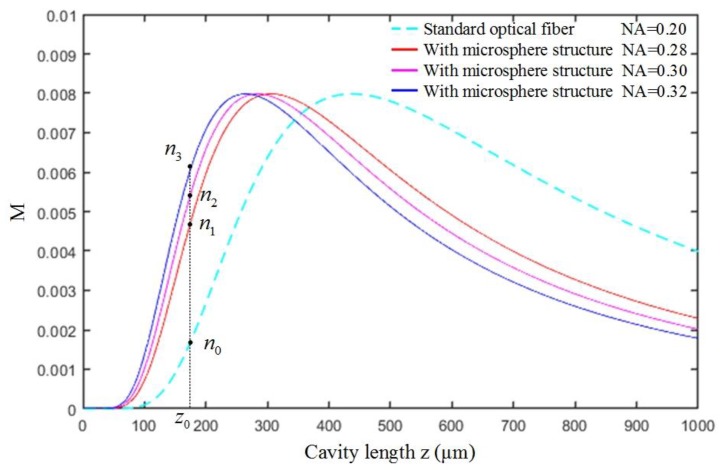
Influence of the numerical aperture on the modulation characteristic curve.

**Figure 3 sensors-20-02233-f003:**
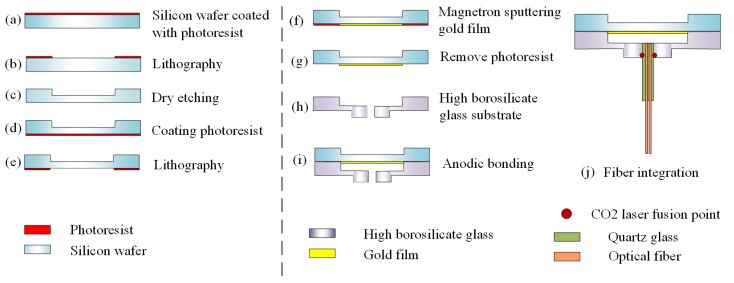
Manufacturing of fiber-optic pressure sensors using MEMS technology.

**Figure 4 sensors-20-02233-f004:**
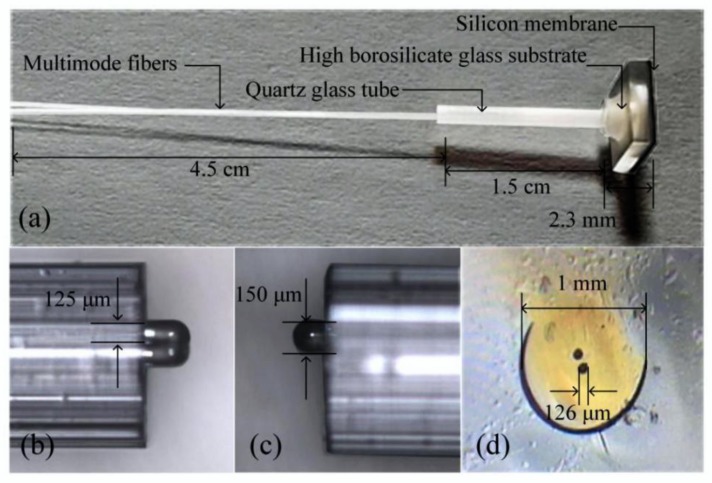
Sensor structure: (**a**) physical drawing of proposed sensor; (**b**) microscopic top view of the microsphere end of optical fiber; (**c**) side view of the optical fiber; and (**d**) sectional view of the double-hole quartz casing.

**Figure 5 sensors-20-02233-f005:**
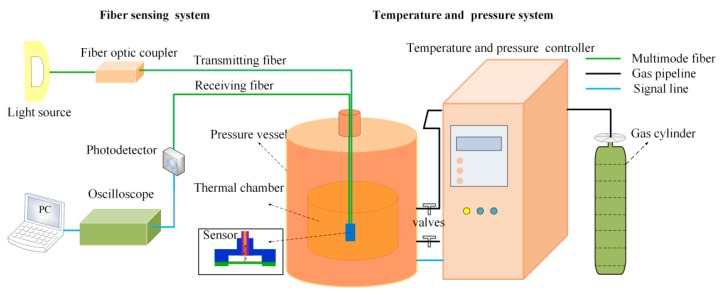
Experimental setup of the sensor for pressure testing under the dynamic temperature.

**Figure 6 sensors-20-02233-f006:**
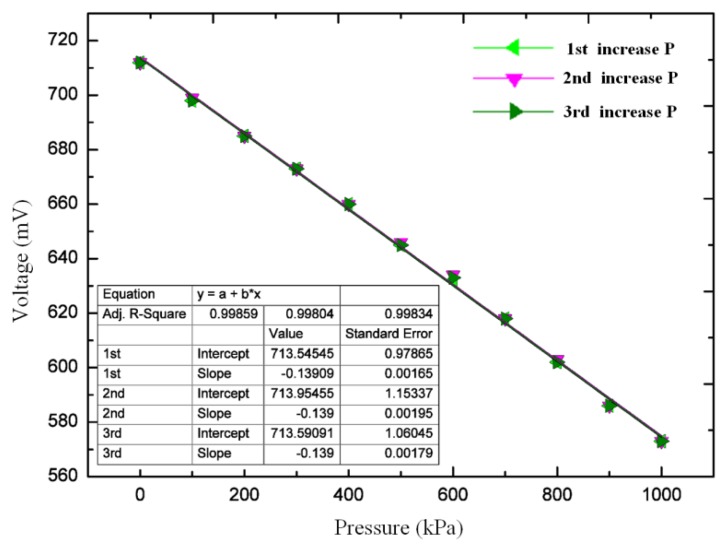
Output voltage versus pressure during three experiments.

**Figure 7 sensors-20-02233-f007:**
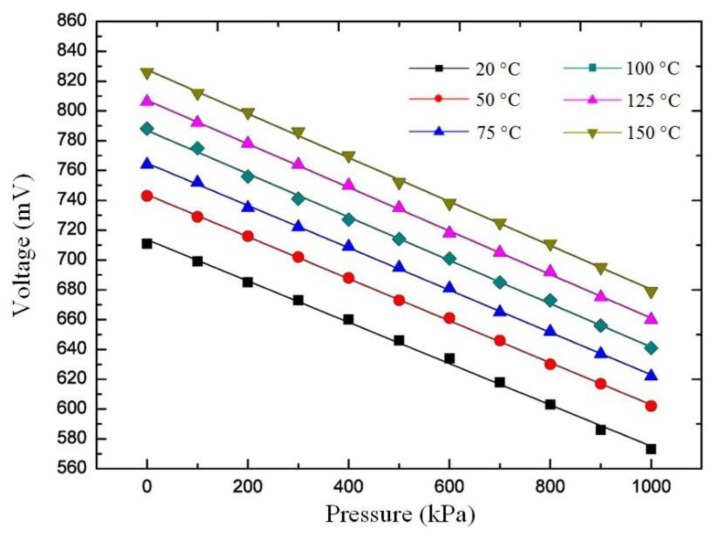
Relationship between the voltage and pressure at different temperatures.

**Figure 8 sensors-20-02233-f008:**
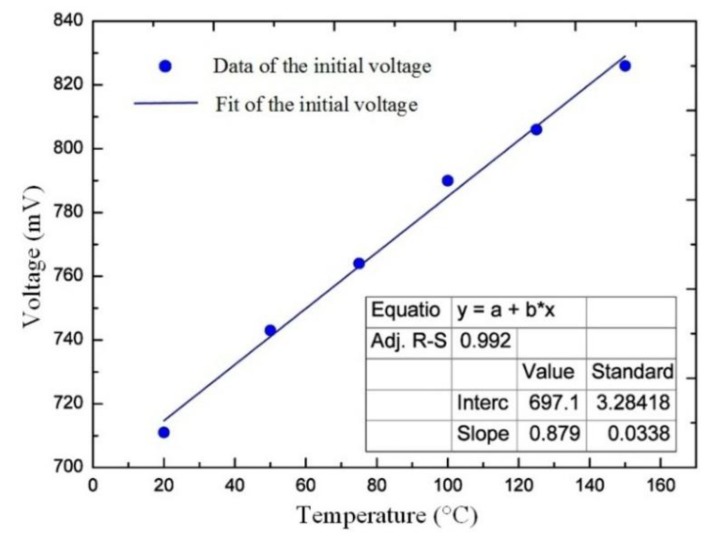
Temperature influence on the initial voltage of the sensor.

**Figure 9 sensors-20-02233-f009:**
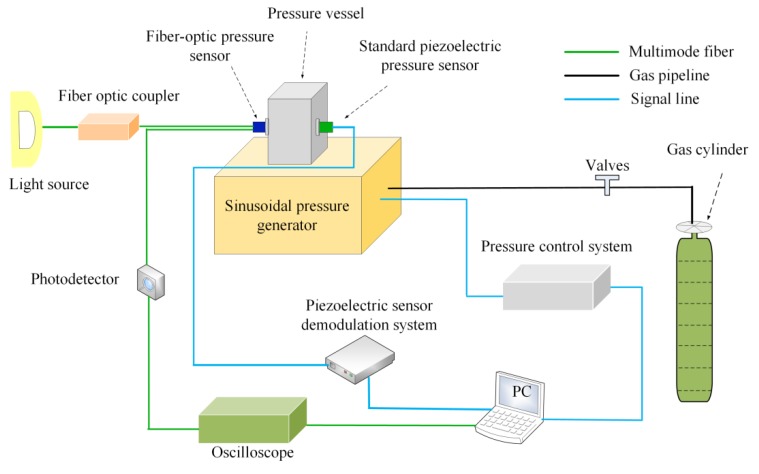
Experimental setup of the dynamic pressure test.

**Figure 10 sensors-20-02233-f010:**
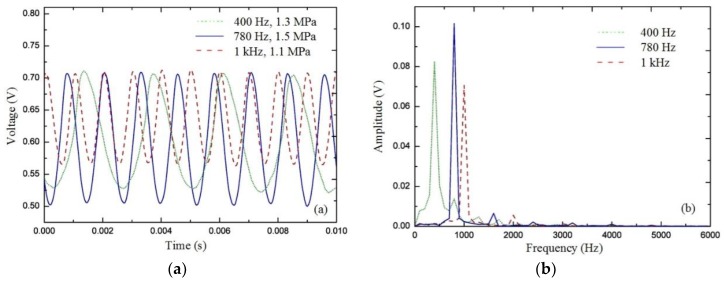
Output of the sensor: (**a**) waveform of voltage and (**b**) fast Fourier transform spectrum of the waveform.

**Table 1 sensors-20-02233-t001:** Structural parameters of the proposed fiber-optic pressure sensor.

Parameter	Symbol	Typical Value (μm)
Core radius of the transmitting fiber	$w_{0}$	31.25
Core radius of the receiving fiber	$w_{r}$	31.25
Center-to-center distance between two fiber cores	s	150

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
