# Peer review of "MEMS-Based Reflective Intensity-Modulated Fiber-Optic Sensor for Pressure Measurements"

_sensors, 2020, doi:10.3390/s20082233_

Round 1

Reviewer 1 Report

Dear Authors,
1. It is necessary to show the novelty of your sensors according to the recent state of the art because the use of the fiber-optic intensity sensors is very well known for pressure measurement.
2. I will recommend extending the introduction part with including the references of the articles which show other configuration of fiber optic sensors with microsphere, such as:
Hirsch M., Listewnik P., Struk P., Weber M., Bechelany M., Szczerska M., ZnO coated fiber optic microsphere sensor for the enhanced refractive index sensing, 2019, Sensors and Actuators A, 298 [10.1016/j.sna.2019.111594]
and
Listewnik P., Hirsch M., Struk P., Weber M., Bechelany M., Jędrzejewska-Szczerska M., Preparation and Characterization of Microsphere ZnO ALD Coating Dedicated for the Fiber-Optic Refractive Index Sensor, 2019, Nanomaterials, 9(2), 306; [doi.org/10.3390/nano9020306]
3. It is necessary to show in detail the procedure of producing the lens on the fiber tips.
4. In my opinion, it is necessary to expand the number of measurements up to 10 which makes the results meaningful from the statistical point of view.

Author Response

Dear reviewer,

Thanks for your review of the manuscript entitled “MEMS-based reflective intensity-modulated fiber-optic sensor for pressure measurements” (Manuscript Number: sensors-755198). We have studied the valuable comments from you carefully, and tried our best to revise the manuscript, and all the revised details are marked in RED color. The main corrections are listed bleow.

Comment 1: It is necessary to show the novelty of your sensors according to the recent state of the art because the use of the fiber-optic intensity sensors is very well known for pressure measurement.

Response: Thank you for your valuable advice.  

Typical optical fiber pressure sensors are mainly consist of two types: intensity modulation and interference. Compared with traditional interferometric fiber optic pressure sensors, the intensity modulation-based pressure sensor presented in this paper has a simpler demodulation system, which is easy to fabricate and low-cost. Due to the intensity demodulation mechanism, the demodulation speed of the intensity-type optical fiber pressure sensor is not limited, so that this type of sensor can test higher frequency dynamic pressure. Because of the limitations of temperature-resistant materials and the current processing technologies, the existing intensity-modulated optical fiber pressures are not suitable for harsh temperature environments or batch-production. Moreover, the sensitivity of the traditional light intensity demodulation pressure sensor is relatively low, which affects the pressure measurement accuracy of the sensor. In this paper, the integrated sensor is allowed a higher sensitivity due to the MEMS technique and the spherical end of the fiber. The sensor was assembled and sealed by using CO2 laser, which is beneficial to improve the sensor performance and avoid thermal mismatch between the adhesive and the fiber-optic. We experimentally tested the temperature and dynamic pressure characteristics of the dual-fiber-based light intensity modulation pressure sensor.

We have made changes in the article.

In this study, we propose a MEMS-based reflective intensity-modulated fiber-optic sensor for pressure measurement. The sensor consists of two multimode optical fibers with a spherical end, a quartz tube with dual holes, a silicon sensitive diaphragm, and a high borosilicate glass substrate (HBGS) integrated by MEMS technique. The sensor is assembled and sealed using CO2 laser, which can withstand harsh environmental pressure tests and the sensor performance is more stable. The temperature features are characterized over the temperature range of 20 °C–150 °C, and the dynamic response are verified at room temperature. The sensitivity of the sensor is significantly improved by fabricating a micro-spherical at the end of the multimode fiber. Due to the intensity mechanism, the sensor has a relatively simple demodulation system and can respond to higher frequency pressure. Besides, the pressure sensor proposed in this paper has the potential to be mass-produced, which can reduce manufacturing costs.

Comment 2: I will recommend extending the introduction part with including the references of the articles which show other configuration of fiber optic sensors with microsphere.

Response: Thanks for the suggestion.

we have extended the introduction part and added relevant references, as shown below: 

With the development of technology, a considerable number of methods have been applied to fabricate the fiber-optic sensors [21–23], for example, MEMS, chemical etching, lasing, applying special optical fibers, and so on [24–28]. Ge et al. proposed an optical MEMS pressure sensor based on a mesa-diaphragm structure [29]. Poeggel et al. reported a femtosecond-laser-based inscription technique for post-fiber-Bragg grating inscription in an extrinsic Fabry–Perot interferometer pressure sensor, the sensor has a small size and high stability [30]. Hirsch et al. reported the fiber-optic microsphere sensors, which realized high-sensitivity refractive index detection [31]. The MEMS-based optical fiber pressure sensors have attracted significant interest, they are quite small and ideal for applications where restricted space or minimal measurement interference is a consideration. 

relevant references:

  1. Listewnik, P.; Hirsch, M.; Struk, P.; Weber, M.; Bechelany, M.; Jędrzejewska-Szczerska, M. Preparation and characterization of microsphere ZnO ALD coating dedicated for the fiber-optic refractive index sensor. Nanomaterials, 2019, 9, 306.
  2. Hirsch, M.; Listewnik, P.; Struk, P.; Weber, M.; Bechelany, M.; Szczerska, M. ZnO coated fiber optic microsphere sensor for the enhanced refractive index sensing. Sens. Actuators, A. 2019, 298.
  3. Song, P.H.; Ma, Z.; Ma, J.; Yang, L.L.; Wei, J.T.; Zhao, Y.M.; Zhang, M.L.; Yang, F.H.; Wang, X.D. Recent progress of miniature MEMS pressure sensors. Micromachines 2020, 11, 56.
  4. Binua, S.; Mahadevan Pillaia, V.P.; Chandrasekaran, N.; Fibre optic displacement sensor for the measurement of amplitude and frequency of vibration. Opt. Laser Technol. 2007, 39, 1537−1543.
  5. Li, H.L.; Deng, H.; Zheng, G.Q.; Shan, M.G.; Zhong, Z.; Liu, B. Reviews on corrugated diaphragms in miniature fiber-optic pressure Sensors. Appl. Sci. 2019, 9, 2241.
  6. Gruca, G.; De Man, S.; Slaman, M.; Rector, J.H.; Iannuzzi, D. Ferrule-top micromachined devices: design, fabrication, performance. Meas. Sci. Technol. 2010, 21, 094033.
  7. Jiang, Y.G.; Li, J.; Zhou, Z.W.; Jiang, X.G.; Zhang, D.Y. Fabrication of all-SiC fiber-optic pressure sensors for high-temperature applications. Sensors. 2016, 16, 1660.
  8. Wang, X.; Wang, S.; Jiang, J.F.; Liu, K.; Zhang, X.Z.; Xiao, M.N.; Xiao, H.; Liu, T.G. Non-destructive residual pressure self-measurement method for the sensing chip of optical Fabry-Perot pressure sensor. Opt. Express. 2017,25, 31937−31947.
  9. Bae, H.; Yun, D.; Liu, H.; Olson, D.A.; Yu, M. Hybrid miniature Fabry–Perot sensor with dual optical cavities for simultaneous pressure and temperature measurements. J. Lightwave Technol. 2014, 32, 85−93.
  10. Ge, Y.X.; Wang, M.; Yan, H.T. Optical MEMS pressure sensor based on a mesa-diaphragm structure. Opt. Express. 2008, 16, 21746−21752.
  11. Poeggel, S.; Duraibabu, D.; Lacraz, A.; Kalli, K.; Tosi, D.; Leen, G.; Lewis, E. Femtosecond-laser-based inscription technique for post-fiber-Bragg grating inscription in an extrinsic Fabry–Perot interferometer pressure sensor. IEEE Sens. J. 2015,16, 396−402.

Comment 3: It is necessary to show in detail the procedure of producing the lens on the fiber tips.

Response: Thank you for your valuable advice.

We have added some descriptions to show in detail the procedure of producing the lens on the fiber tips. We have made changes in the paper, as shown below:

Microsphere structure was made by laser heating at the end of the fiber. The CO2 laser fusion splicer (LZM–110, Fujikura, Ltd, Tokyo, Japan) was used in the production process. The laser power was 175 bits (about 5.2W), the laser heating time was 2 s, and the heating was performed twice [37].

relevant references:

37. Guzowski, B.; Lisik, Z.; Tosik, G. Realization of optical fibers terminated with ball lenses. Bull. Pol. Acad. Sci. Tech. Sci. 2015, 64, 279−282.

Comment 4: In my opinion, it is necessary to expand the number of measurements up to 10 which makes the results meaningful from the statistical point of view.

Response: Thank you for your valuable advice.

We have added the experimental datas(under 75 °C , 125 °C),as shown in the Word file. 

in Figure 7. Relationship between the voltage and pressure at different temperatures.

Reviewer 2 Report

The work describes a fiber optic pressure sensor, fabricated with a MEMS structure. The work is nice and the characterization is well done; I only have minor comments:

(1) Introduction: please cite also other MEMS-integrated pressure sensors, e.g.:

  • Gruca G, De Man S, Slaman M, Rector JH, Iannuzzi D. Ferrule-top micromachined devices: design, fabrication, performance. Measurement Science and Technology. 2010 Jul 28;21(9):094033

and Fabry-Perot interferometers:

  • Poeggel S, Tosi D, Fusco F, Ippolito J, Lupoli L, Mirone V, Sannino S, Leen G, Lewis E. Fiber-optic EFPI pressure sensors for in vivo urodynamic analysis. IEEE Sensors Journal. 2014 Mar 11;14(7):2335-40.
  • Yu Q, Zhou X. Pressure sensor based on the fiber-optic extrinsic Fabry-Perot interferometer. Photonic Sensors. 2011 Mar 1;1(1):72-83.

It i important to note that some of these works also integrate the FP interferometer with an FBG, to make dual temperature-pressure detection:

  • Tosi D, Macchi EG, Braschi G, Cigada A, Gallati M, Rossi S, Poeggel S, Leen G, Lewis E. Fiber-optic combined FPI/FBG sensors for monitoring of radiofrequency thermal ablation of liver tumors: ex vivo experiments. Applied optics. 2014 Apr 1;53(10):2136-44.
  • Bae H, Yun D, Liu H, Olson DA, Yu M. Hybrid miniature Fabry–Perot sensor with dual optical cavities for simultaneous pressure and temperature measurements. Journal of Lightwave Technology. 2014 Apr 15;32(8):1585-93.
  • Poeggel S, Duraibabu D, Lacraz A, Kalli K, Tosi D, Leen G, Lewis E. Femtosecond-laser-based inscription technique for post-fiber-Bragg grating inscription in an extrinsic Fabry–Perot interferometer pressure sensor. IEEE Sensors Journal. 2015 May 18;16(10):3396-402.

(2) Figure 2: how was data generated? Did the authors adapt the theoretical Faria's fiber bundle model (Faria JB. A theoretical analysis of the bifurcated fiber bundle displacement sensor. IEEE Transactions on instrumentation and measurement. 1998 Jun;47(3):742-7), or did they use a numerical methods (e.g. Tosi D, Perrone G, Vallan A. Performance analysis of a noncontact plastic fiber optical fiber displacement sensor with compensation of target reflectivity. Journal of Sensors. 2013;2013) adapting for the NA and fiber geometry?

(3) Figure 7. Please report the slope for each curve, so to ensure that the sensitivity is not affected by the temperature. The authors can list this information here instead of Figure 9, which is a bit redundant.

Author Response

Dear reviewer,

Thanks for your review of the manuscript entitled “MEMS-based reflective intensity-modulated fiber-optic sensor for pressure measurements” (Manuscript Number: sensors-755198). We have studied the valuable comments from you carefully, and tried our best to revise the manuscript, and all the revised details are marked in RED color. The main corrections are listed bleow.

Comment 1: Please cite also other MEMS-integrated pressure sensors, Fabry-Perot interferometers, It is important to note that some of  these works also integrate the FP interferometer with an FBG, to make dual temperature-pressure detection.

Response: Thank you for your valuable advice.

We have added relevant references and we have made changes in the paper, as shown below:

With the development of technology, a considerable number of methods have been applied to fabricate the fiber-optic sensors [21–23], for example, MEMS, chemical etching, lasing, applying special optical fibers, and so on [24–28]. Ge et al. proposed an optical MEMS pressure sensor based on a mesa-diaphragm structure [29]. Poeggel et al. reported a femtosecond-laser-based inscription technique for post-fiber-Bragg grating inscription in an extrinsic Fabry–Perot interferometer pressure sensor, the sensor has a small size and high stability [30]. Hirsch et al. reported the fiber-optic microsphere sensors, which realized high-sensitivity refractive index detection [31]. The MEMS-based optical fiber pressure sensors have attracted significant interest, they are quite small and ideal for applications where restricted space or minimal measurement interference is a consideration.   

References:

24.Gruca, G.; De Man, S.; Slaman, M.; Rector, J.H.; Iannuzzi, D. Ferrule-top micromachined devices: design, fabrication, performance. Meas. Sci. Technol. 2010, 21, 094033.

30.Poeggel, S.; Duraibabu, D.; Lacraz, A.; Kalli, K.; Tosi, D.; Leen, G.; Lewis, E. Femtosecond-laser-based inscription technique for post-fiber-Bragg grating inscription in an extrinsic Fabry–Perot interferometer pressure sensor. IEEE Sens. J. 2015,16, 396−402.

27.Bae, H.; Yun, D.; Liu, H.; Olson, D.A.; Yu, M. Hybrid miniature Fabry–Perot sensor with dual optical cavities for simultaneous pressure and temperature measurements. J. Lightwave Technol. 2014, 32, 85−93.

21.Song, P.H.; Ma, Z.; Ma, J.; Yang, L.L.; Wei, J.T.; Zhao, Y.M.; Zhang, M.L.; Yang, F.H.; Wang, X.D. Recent progress of miniature MEMS pressure sensors. Micromachines 2020, 11, 56.

22.Binua, S.; Mahadevan Pillaia, V.P.; Chandrasekaran, N.; Fibre optic displacement sensor for the measurement of amplitude and frequency of vibration. Opt. Laser Technol. 2007, 39, 1537−1543.

23.Li, H.L.; Deng, H.; Zheng, G.Q.; Shan, M.G.; Zhong, Z.; Liu, B. Reviews on corrugated diaphragms in miniature fiber-optic pressure Sensors. Appl. Sci. 2019, 9, 2241.

25.Jiang, Y.G.; Li, J.; Zhou, Z.W.; Jiang, X.G.; Zhang, D.Y. Fabrication of all-SiC fiber-optic pressure sensors for high-temperature applications. Sensors. 2016, 16, 1660.

26.Wang, X.; Wang, S.; Jiang, J.F.; Liu, K.; Zhang, X.Z.; Xiao, M.N.; Xiao, H.; Liu, T.G. Non-destructive residual pressure self-measurement method for the sensing chip of optical Fabry-Perot pressure sensor. Opt. Express. 2017,25, 31937−31947.

28.Listewnik, P.; Hirsch, M.; Struk, P.; Weber, M.; Bechelany, M.; Jędrzejewska-Szczerska, M. Preparation and characterization of microsphere ZnO ALD coating dedicated for the fiber-optic refractive index sensor. Nanomaterials, 2019, 9, 306.

29.Ge, Y.X.; Wang, M.; Yan, H.T. Optical MEMS pressure sensor based on a mesa-diaphragm structure. Opt. Express. 2008, 16, 21746−21752.

31.Hirsch, M.; Listewnik, P.; Struk, P.; Weber, M.; Bechelany, M.; Szczerska, M. ZnO coated fiber optic microsphere sensor for the enhanced refractive index sensing. Sens. Actuators, A. 2019, 298.

Comment 2: how was data generated? Did the authors adapt the theoretical Faria's fiber bundle model (Faria JB. A theoretical analysis of the bifurcated fiber bundle displacement sensor. IEEE Transactions on instrumentation and measurement. 1998 Jun;47(3):742-7), or did they use a numerical methods (e.g. Tosi D, Perrone G, Vallan A. Performance analysis of a noncontact plastic fiber optical fiber displacement sensor with compensation of target reflectivity. Journal of Sensors. 2013;2013) adapting for the NA and fiber geometry?

Response: Thank you for your valuable advice.

We adapt the theoretical Faria's fiber bundle model (Faria JB. A theoretical analysis of the bifurcated fiber bundle displacement sensor. IEEE Transactions on instrumentation and measurement. 1998 Jun;47(3):742-7). ( Formula part are shown in the Word file)

According to references[1,2], When performing specific calculations on the above theory, the light emitted from the end face of the emitting fiber can be regarded as a geometric beam, the half-width of the Gaussian beam can be expressed as:(Formula part are shown in the Word file)

According to references[3–5], The optical fiber with the microsphere structure has a higher numerical aperture (NA). And NA is related to the radius of curvature of the spherical fiber end:(Formula part is shown in the Word file)

Where nlens is the refractive index of the lens, m is diameter of the light spot and P is radius of the lens.

references:

[1] Perret, L.; Chassagne, L.; Topcu, S.; Ruaux, P.; Cagneau, B.; Alayli, Y. Fiber optics sensor for sub-nanometric displacement and wide bandwidth systems. Sensors Actuators A 2011, 165, 189-193.

[2] Golnabi, H.; Azimi, P. Design and operation of a double-fiber displacement sensor.  Optics Communications 281 (2008) 614–620

[3]Guzowski, B.; Lisik, Z.; Tosik, G. Realization of optical fibers terminated with ball lenses. Bull. Pol. Acad. Sci. Tech. Sci. 2015, 64, 279-282.

[4]C. A. Brackett. On the efficiency of coupling light from stripe-geometry GaAs lasers into multimode optical fibers.

[5]Guzowski, B.; Lakomski, M. Realization of fiber optic displacement sensors.

Comment 3:  Please report the slope for each curve, so to ensure that the sensitivity is not affected by the temperature. The authors can list this information here instead of Figure 9, which is a bit redundant.

Response: Thank you for your valuable advice.

 We have made changes in the paper, as shown below:

The pressure sensitivities at temperatures of 20 °C, 50 °C, 75 °C, 100 °C, 125 °C and 150 °C, were 0.1387, 0.1409, 0.1431, 0.1452, 0.1460, 0.1473 mV/kPa, respectively. It can be seen that the temperature has a small effect on the sensitivity of the sensor, which may be caused by complex physical mechanisms such as the elastic modulus of the sensitive diaphragm changes with the temperature.

Reviewer 3 Report

This paper proposes a reflective intensity-modulated fiber-optic pressure sensor based on microelectromechanical systems.

The authors present an extensive list of references covering many types of pressures sensors but there are few references about the specific technology described on the paper.

There many references on the technical literature describing similar sensors, such as:

  • Fibre optic displacement sensor for the measurement of amplitude and frequency of vibration - https://doi.org/10.1016/j.optlastec.2006.12.008
  • Reviews on Corrugated Diaphragms in Miniature Fiber-Optic Pressure Sensors - https://doi.org/10.3390/app9112241
  • Recent Progress of Miniature MEMS Pressure Sensors - https://doi.org/10.3390/mi11010056
  • Non-destructive residual pressure self measurement method for the sensing chip of
  • optical Fabry-Perot pressure sensor - https://doi.org/10.1364/OE.25.030939
  • Optical MEMS pressure sensor based on a mesa-diaphragm structure -https://doi.org/10.1364/OE.16.021746  
  • Fabrication of All-SiC Fiber-Optic Pressure Sensors for High-Temperature Applications - https://doi.org/10.3390/s16101660

In addition it was not possible to detect what is the novelty of this paper. The authors must clarify this issue.

In the experiments the author must comment:

  • The maximum temperature tested was 150°C. What is the limitation of the sensor application in temperatures higher than 150°C?
  • The dynamic response of the sensor was tested just in one single frequency (780 Hz). Why the sensor was not tested in a wide band frequency spectrum?

Author Response

Dear reviewer,

Thanks for your review of the manuscript entitled “MEMS-based reflective intensity-modulated fiber-optic sensor for pressure measurements” (Manuscript Number: sensors-755198). We have studied the valuable comments from you carefully, and tried our best to revise the manuscript, and all the revised details are marked in RED color. The main corrections are listed bleow.

Comment 1: The authors present an extensive list of references covering many types of pressures sensors but there are few references about the specific technology described on the paper. 

Response: Thanks for the suggestion.

we have added descriptionsabout the specific technology and relevant references in the paper, as shown below:

With the development of technology, a considerable number of methods have been applied to fabricate the fiber-optic sensors [21–23], for example, MEMS, chemical etching, lasing, applying special optical fibers, and so on [24–28]. Ge et al. proposed an optical MEMS pressure sensor based on a mesa-diaphragm structure [29]. Poeggel et al. reported a femtosecond-laser-based inscription technique for post-fiber-Bragg grating inscription in an extrinsic Fabry–Perot interferometer pressure sensor, the sensor has a small size and high stability [30]. Hirsch et al. reported the fiber-optic microsphere sensors, which realized high-sensitivity refractive index detection [31]. The MEMS-based optical fiber pressure sensors have attracted significant interest, they are quite small and ideal for applications where restricted space or minimal measurement interference is a consideration.  

References:

21.Song, P.H.; Ma, Z.; Ma, J.; Yang, L.L.; Wei, J.T.; Zhao, Y.M.; Zhang, M.L.; Yang, F.H.; Wang, X.D. Recent progress of miniature MEMS pressure sensors. Micromachines 2020, 11, 56.

22.Binua, S.; Mahadevan Pillaia, V.P.; Chandrasekaran, N.; Fibre optic displacement sensor for the measurement of amplitude and frequency of vibration. Opt. Laser Technol. 2007, 39, 1537−1543.

23.Li, H.L.; Deng, H.; Zheng, G.Q.; Shan, M.G.; Zhong, Z.; Liu, B. Reviews on corrugated diaphragms in miniature fiber-optic pressure Sensors. Appl. Sci. 2019, 9, 2241.

29.Ge, Y.X.; Wang, M.; Yan, H.T. Optical MEMS pressure sensor based on a mesa-diaphragm structure. Opt. Express. 2008, 16, 21746−21752.

25.Jiang, Y.G.; Li, J.; Zhou, Z.W.; Jiang, X.G.; Zhang, D.Y. Fabrication of all-SiC fiber-optic pressure sensors for high-temperature applications. Sensors. 2016, 16, 1660.

26.Wang, X.; Wang, S.; Jiang, J.F.; Liu, K.; Zhang, X.Z.; Xiao, M.N.; Xiao, H.; Liu, T.G. Non-destructive residual pressure self-measurement method for the sensing chip of optical Fabry-Perot pressure sensor. Opt. Express. 2017,25, 31937−31947.

24.Gruca, G.; De Man, S.; Slaman, M.; Rector, J.H.; Iannuzzi, D. Ferrule-top micromachined devices: design, fabrication, performance. Meas. Sci. Technol. 2010, 21, 094033.

27.Bae, H.; Yun, D.; Liu, H.; Olson, D.A.; Yu, M. Hybrid miniature Fabry–Perot sensor with dual optical cavities for simultaneous pressure and temperature measurements. J. Lightwave Technol. 2014, 32, 85−93.

28.Listewnik, P.; Hirsch, M.; Struk, P.; Weber, M.; Bechelany, M.; Jędrzejewska-Szczerska, M. Preparation and characterization of microsphere ZnO ALD coating dedicated for the fiber-optic refractive index sensor. Nanomaterials, 2019, 9, 306.

30.Poeggel, S.; Duraibabu, D.; Lacraz, A.; Kalli, K.; Tosi, D.; Leen, G.; Lewis, E. Femtosecond-laser-based inscription technique for post-fiber-Bragg grating inscription in an extrinsic Fabry–Perot interferometer pressure sensor. IEEE Sens. J. 2015,16, 396−402.

31.Hirsch, M.; Listewnik, P.; Struk, P.; Weber, M.; Bechelany, M.; Szczerska, M. ZnO coated fiber optic microsphere sensor for the enhanced refractive index sensing. Sens. Actuators, A. 2019, 298.

Comment 2: In addition it was not possible to detect what is the novelty of this paper. The authors must clarify this issue.

Response: Thanks for the suggestion.

Typical optical fiber pressure sensors are mainly consist of two types: intensity modulation and interference. Compared with traditional interferometric fiber optic pressure sensors, the intensity modulation-based pressure sensor presented in this paper has a simpler demodulation system, which is easy to fabricate and low-cost. Due to the intensity demodulation mechanism, the demodulation speed of the intensity-type optical fiber pressure sensor is not limited, so that this type of sensor can test higher frequency dynamic pressure. Because of the limitations of temperature-resistant materials and the current processing technologies, the existing intensity-modulated optical fiber pressures are not suitable for harsh temperature environments or batch-production. Moreover, the sensitivity of the traditional light intensity demodulation pressure sensor is relatively low, which affects the pressure measurement accuracy of the sensor. In this paper, the integrated sensor is allowed a higher sensitivity due to the MEMS technique and the spherical end of the fiber. The sensor was assembled and sealed by using CO2 laser, which is beneficial to improve the sensor performance and avoid thermal mismatch between the adhesive and the fiber-optic. We experimentally tested the temperature and dynamic pressure characteristics of the dual-fiber-based light intensity modulation pressure sensor.

We have added some related descriptions in the article.

In this study, we propose a MEMS-based reflective intensity-modulated fiber-optic sensor for pressure measurement. The sensor consists of two multimode optical fibers with a spherical end, a quartz tube with dual holes, a silicon sensitive diaphragm, and a high borosilicate glass substrate (HBGS) integrated by MEMS technique. The sensor is assembled and sealed using CO2 laser, which can withstand harsh environmental pressure tests and the sensor performance is more stable. The temperature features are characterized over the temperature range of 20 °C–150 °C, and the dynamic response are verified at room temperature. The sensitivity of the sensor is significantly improved by fabricating a micro-spherical at the end of the multimode fiber. Due to the intensity mechanism, the sensor has a relatively simple demodulation system and can respond to higher frequency pressure. Besides, the pressure sensor proposed in this paper has the potential to be mass-produced, which can reduce manufacturing costs.

Comment 3:

The maximum temperature tested was 150°C. What is the limitation of the sensor application in temperatures higher than 150°C ?

Response: Thank you for your valuable advice.

During the experiment, when the temperature exceeded 150 °C at a pressure of 1 MPa, the welding part of the sensor was damaged and resulting in the pressure chamber of the sensor not being sealed. During the laser welding process, the sensor will be damaged when the laser power is too high. If the laser power is too low, the sensor will not be sealed. The laser power parameters need to be further optimized to adapt to higher temperature environments, which will be our next research.

Comment 4: The dynamic response of the sensor was tested just in one single frequency (780 Hz). Why the sensor was not tested in a wide band frequency spectrum?

Response: Thanks for the suggestion.

This article is to verify the pressure response of the sensor under dynamic conditions, so only experiments performed at   400 Hz, 780 Hz, and 1 kHz . The experimental results are shown in the figures 1-3.(show in the Word profile)

Figure 1. Output of the sensor under 400 Hz pressure, at room temperature: (a) waveform of pressure and (b) fast Fourier transform spectrum of the waveform.

Figure 2. Output of the sensor under 780 Hz pressure, at room temperature: (a) waveform of pressure and (b) fast Fourier transform spectrum of the waveform.

Figure 3. Output of the sensor under 1 kHz pressure, at room temperature: (a) waveform of pressure and (b) fast Fourier transform spectrum of the waveform.

The pressure source generated by the pressure generator is unstable during lower frequency tests, so the experiments are not included in the text. We have added the experimental data in the article at 1 kHz frequency as blew:(figures shown in the Word file)

Figure 11. Output of the sensor: (a) waveform of pressure and (b) fast Fourier transform spectrum of the waveform.

Round 2

Reviewer 1 Report

Dear Authors,

I recommend this manuscript to be accepted.

Author Response

Dear reviewer,

Thanks for your review of the manuscript entitled “MEMS-based reflective intensity-modulated fiber-optic sensor for pressure measurements” (Manuscript Number: sensors-755198). Thank you for your valuable advice.

Reviewer 3 Report

The authors answered the major questions.

Some few corrections must be provided: the authors must write in the manuscript the reasons for the limiting temperature of 150°C and insert also the results or comments for frequency testing in 400 Hz.

Author Response

Dear reviewer,

Thanks for your review of the manuscript entitled “MEMS-based reflective intensity-modulated fiber-optic sensor for pressure measurements” (Manuscript Number: sensors-755198). We have studied the valuable comments from you carefully, and tried our best to revise the manuscript, and all the revised details are marked in RED color. The main corrections are listed bleow.

Comment 1: The authors must write in the manuscript the reasons for the limiting temperature of 150°C.

Response: Thank you for your valuable advice.

We have written the reasons for the limiting temperature of 150°C in the manuscript.

When the temperature exceeded 150 °C at a pressure of 1 MPa, the welding part of the sensor was damaged, resulting in the pressure cavity of the sensor not being sealed. Therefore, the sensor can operate at a higher temperature than 150 °C, when the welding process is optimized.

Comment 2: The authors must insert also the results or comments for frequency testing in 400 Hz.

Response: Thank you for your valuable advice.

We have made changes in the article.

To verify the pressure response of the sensor under dynamic conditions, experiments at 400 Hz, 780 Hz, and 1 kHz were performed.

Figure 10. Output of the sensor: (a) waveform of voltage and (b) fast Fourier transform spectrum of the waveform. (figure 10 shown in the Word file)
